# Simulated and Verification of Mass and Heat Transfer Coupled Model of Jujube Slices Dried by Hot Air Combined with Radio Frequency Heat Treatment at Different Drying Stages

**DOI:** 10.3390/foods12163025

**Published:** 2023-08-11

**Authors:** Shuaitao Cao, Chenyan Yang, Yongzhen Zang, Yang Li, Jiangwei Gu, Haiyang Ding, Xuedong Yao, Rongguang Zhu, Qiang Wang, Wancheng Dong, Yong Huang

**Affiliations:** 1College of Mechanical and Electrical Engineering, Shihezi University, Shihezi 832003, China; cao12307@163.com (S.C.);; 2Key Laboratory of Northwest Agricultural Equipment, Ministry of Agriculture and Rural Affairs, Shihezi 832003, China; 3Key Laboratory of Modern Agricultural Machinery Corps, Shihezi 832003, China

**Keywords:** winter jujube slice, hot air drying, radio-frequency heat treatment, numerical models, uniformity

## Abstract

This study investigates the impact of radio frequency (RF) heat treatment on heat and mass transfer during the hot air drying of jujube slices. Experiments were conducted at different drying stages, comparing single-hot air drying with hot air combined with RF treatment. Numerical models using COMSOL Multiphysics^®^ were developed to simulate the process, and the results were compared to validate the models. The maximum difference between the simulated value of the center temperature and the experimental value was 6.9 °C, while the minimum difference was 0.1 °C. The maximum difference in average surface temperature was 1.7 °C, with a minimum of 0.3 °C. The determination coefficient (R2) between the simulated experimental values of HA and the early (E-HA + RF), middle (M-HA + RF), and later (L-HA + RF) groups was 0.964, 0.987, 0.961, and 0.977, respectively. The study demonstrates that RF treatment reduces drying time, enhances internal temperature, promotes consistent heat and mass transfer, and accelerates moisture diffusion in jujube slices. Furthermore, the later the RF treatment is applied, the greater the increase in internal temperature and the faster the decrease in moisture content. This research elucidates the mechanism by which RF heat treatment influences heat transfer in hot air-dried jujube slices.

## 1. Introduction

The Jujube (Ziziphus jujuba Mill), native to China. As the biggest creation base in China and even on the planet, Xinjiang represents over half of the homegrown jujube production (Department of Far-reaching Measurements of the Public Economy, 2022). There has been a consistent expansion in the need for jujube for its rich healthful substances such as proteins, dietary fiber, ascorbic corrosive, and jujube polysaccharide [1]. Jujube samples can be eaten fresh, but consumers prefer the dried jujube slice, a by-product of jujube with a long shelf life and a distinctive flavor. The dried jujube can be consumed as compote, dried fruit, or as an addition to soup.

Hot air drying is a popular dehydration technique for jujube slices because of its low cost and ease of usage. Nevertheless, it must consider the oxidation-browning response and loss of heat-sensitive nutrients [2]. Hot air drying in combination with other methods has been investigated to meet consumer demands. It is typical to use radio frequency (RF) combined with hot air drying [3,4,5]. Hot air drying can effectively remove the surface moisture of the jujube slices, while radio frequency heating rapidly diffuses the moisture inside the jujube slices to the surface due to its unique pressure gradient [6]. During the RF process, heat is mainly generated inside the jujube slices. With the pressure gradient as the main driving force, the internal water of the jujube slices diffuses to the surface [7].

Wang studied the influence of radio frequency heat treatment on mid-short-wave infrared drying dynamics and the quality characteristics of red dates. It was found that the drying time of the radio frequency heat treatment group was shortened by 21.2%~29.3% compared with the control group, and the quality of dry products was better [8]. Zhou compared and analyzed several drying methods about heating uniformity, efficiency, and quality of kiwifruit slices dried by radio frequency vacuum drying and hot air drying and found that the drying time of radio frequency vacuum drying technology was reduced by 65% compared with hot air drying (60 °C) [9].

Whether it is single hot air drying or RF combined with hot air drying, its essence is the transfer of moisture and heat. The uneven temperature distribution in the drying process may cause great damage to the structure of the material, thus affecting the texture characteristics of the material and making it taste worse [10]. In addition, in the drying process, too low temperatures in the cold area will easily cause the residue of harmful microorganisms, while too high temperatures in the hot area will cause the loss of heat-sensitive nutrients or the phenomenon of regional scorching, thus producing harmful substances for the human body [11]. Uniform moisture distribution means that dry products have a relatively complete and uniform microstructure. Dry products with a better microstructure have better quality, retain more nutrients, and have more ideal texture characteristics [12]. The moisture content of the product affects its moisture activity, which in turn is related to the rate of oxidative reactions and non-enzymatic browning. Phytochemical degradation occurred more rapidly at higher moisture contents [13]. Therefore, moisture distribution is of great significance for product quality and storage.

In drying experiments, it is difficult to visualize the effects of different factors on the product’s interior, such as temperature and moisture concentration [14]. Additionally, experimental procedures have several drawbacks, including lengthy cycles, high economic costs, poor efficiency, and operational challenges. Computer simulation is a more practical, adaptable, and easy technique to disclose the hidden characteristics of items during a complicated drying process than the experimental approach [15,16,17,18,19,20,21,22]. The effectiveness of computer modeling in predicting and adjusting process parameters for RF heating has already been shown [23]. In order to reveal the influence mechanism of RF heat treatment on the heat and mass transfer of hot air drying jujube slices, it is crucial to develop mathematical models of single hot air drying and RF hot air combined drying.

As shown in Figure 1, a mathematical model of single hot air drying and hot air combined radio frequency drying of red date slices at different drying stages was established. The model parameters were determined, and the basic assumptions of the model were made. The drying process was simulated with COMSOL 6.0 software. Through drying experiments, the model was verified by the center temperature, surface temperature, and moisture content.

## 2. Materials and Methods

### 2.1. Preparation of Sample

The winter dates used in this test were purchased from the local wholesale market (Shihezi, Xinjiang). High-quality winter dates with a regular appearance, no mechanical damage, and consistent color and size were selected as experimental materials. The surface of the selected winter dates was cleaned and then put into a sealed bag. Put the above winter dates in a freezer with a temperature of 4 °C and refrigerate them for 24 h for use. The initial moisture content of winter jujube was determined by the oven method, and the value was 79.6 ± 1% [24].

Before the experiment began, the refrigerated winter dates were removed from the freezer, and they were cleaned and dried after the temperature reached room temperature, then removed the kernel and cut them lengthwise at 8 ± 0.5 mm. Sealed the jujube slices, put them in a constant temperature and humidity freezer for 24 h for the later experiment.

### 2.2. Experimental Equipment

The experimental equipment used in this study is shown in Table 1.

**Table 1 foods-12-03025-t001:** Experimental equipment.

Equipment	Model and Manufacturer
RF drying system (equipped with hot air drying system)	SO-6F (frequency of 27.12 MHz, rated output power of 6 kW), Monga Strayfield Private Limited, Monga, India. As shown in Figure 2
electronic balance	BSM220.4,Shanghai Zhuojing Electronic Technology Co., LTD., Shanghai, China)
fiber optic thermometer	HQ-FTS-I9C01, Xi ‘an Herch Opto Electronic Technology Co., LTD., Xi’an, China)
freezer	BCD-267G, Hisense Rongsheng Freezer Co., LTD., Foshan, China)
differential scanning calorimeter	DSC214 Polyma, NETZSCH, Bavaria, Germany
thermal characteristics analyzer	KD2pro, Beijing Gaolitei Technology Co., LTD., Beijing, China
terminal open-circuit coaxial probe dielectric characteristics measurement system	E4991B, Keysight Technologies Malaysia Sdn Bhd, Penang, Malaysia. As shown in Figure 3
water bath kettle	RC-HH-2, Beijing Ruicheng Yongchuang Technology Co., LTD., Beijing, China
infrared camera	RSE 600, Fluke Corporation, Washington, P.O. Box 9090 Everett, WA 98206-9090 U.S.A.

### 2.3. Establishment of Mathematical Models

#### 2.3.1. Mass and Heat Transfer

Fick’s diffusion law explores how the internal moisture of a material changes over time. The shape change of the material due to shrinkage stress is ignored in the drying process. It is assumed that the moisture inside the material is only transferred and diffused in the form of liquid during the drying process. The driving force of water diffusion is the concentration gradient of liquid moisture. The heat and mass transfer equation in this study can be expressed as the heat and mass balance equation based on the Fourier equation and Fick’s second diffusion law, as shown in Equations (1) and (2) [25,26]: (1)$$\rho_{s}C_{p}\frac{\partial T}{\partial t}+\rho_{s}C_{p}\mu \nabla T=(k\nabla T)+Q_{rf}$$
(2)$$\frac{\partial C}{\partial t}+\nabla \cdot \left(-D_{eff}\nabla C\right)=0$$
where *ρ*_s_ is the density of jujube slice (kg/m^3^); *C_p_* is the specific heat of jujube slice (J·kg^−1^·K^−1^); *T* is the temperature at time *t* (°C); *k* is the thermal conductivity of the jujube slice (W·m^−1^·K^−1^); *μ* is the dynamic viscosity of air; *Q_rf_* is the volume of the radio frequency heat source (W/m^3^); *c* is the moisture concentration (kg/m^3^); *D_eff_* is the effective moisture diffusivity (m^2^/s); *t* is the drying time (s).

#### 2.3.2. Governing Equations of the RF Heating Process

The volume of the RF heat source (*Q_rf_*, W/m^3^) refers to the conversion power of electromagnetic energy into heat energy, which is related to the electric field strength, frequency, and dielectric characteristics of the material. Under the premise of determining the radio frequency system and electric field strength, *Q_rf_* can be obtained by Equation (3) [27]:(3)$$Q_{rf}=2\pi f\varepsilon_{0}\varepsilon^{\prime \prime }|\vec{E}{|}^{2}=\omega \varepsilon_{0}\varepsilon^{\prime \prime }|\vec{E}{|}^{2}$$

In the formula, *f* is the frequency (27.12 MHz), *ε*_0_ is the vacuum permittivity (8.854 × 10^−12^ F/m), *ε*” is the imaginary part of the complex number relative permittivity, namely the dielectric loss factor, and $\left|\vec{E}\right|$ is the modulus of electric field *E*, and $\vec{E}=-\vec{\nabla }V$, Equation (3) can be converted into a simpler Laplace equation for calculation [28]. The calculation formula is as follows (4):(4)$$-\vec{\nabla }\cdot \left(\left(\sigma +j2\pi f\varepsilon_{0}\varepsilon^{\prime }\right)\vec{\nabla }V\right)=0$$
where *σ* is the electrical conductivity of the food material (S·m^−1^); *ε*’ is the relative dielectric constant of the food material (-); and *V* is the electric potential across the electrode gap (*V*).

The electric field at any point inside the RF heating cavity can be solved by Equation (4). The upper plate is set as an electromagnetic source due to the introduction of high-frequency electromagnetic energy, and the lower plate is set to ground (*V* = *O*). The upper plate voltage will change with the change in plate spacing. In this study, the upper plate voltage is estimated by the above equation. The values are then fine-tuned using trial and error until the predicted results match the test results of the sample (temperature and moisture content changes). Finally, the distance between the plates was determined to be 110 mm.

#### 2.3.3. Boundary Equation

Assuming that evaporation and convection occur at the boundary of the jujube slice, the boundary equation of heat and mass transfer equations in the computational domain can be expressed by Equations (5) and (6) [29]:(5)$$-(k\nabla T)=h_{T}\left(T_{\mathrm{air}}-T_{\infty }\right)+h_{m}\rho \left(M-M_{e}\right)h_{fg}$$
(6)$$\left(-D_{eff}\nabla C\right)=h_{m}\left(C_{b}-C\right)$$
where *h_T_* is heat transfer coefficient (W/(m^2^·K)); *T*_air_ is dry air temperature (°C), *Me* is equilibrium moisture content (kg/kg, d.b.), and *h_m_* is mass transfer coefficient (m/s). *h_fg_* is the latent heat of evaporation (J/kg); *C_b_* is the overall moisture concentration (kg/m^3^). 

### 2.4. Determination of Model Parameters

#### 2.4.1. Specific Heat Capacity

Specific heat capacity is one of the important thermal physical parameters of jujube slices, and its value is related to the temperature, moisture content, texture properties, and other factors of jujube slices. In this study, a DSC214 differential scanning calorimeter was used to measure the specific heat capacity of four jujube slices with different moisture content, which has the advantages of fast speed, a small amount of required samples, accuracy, and high precision.

Sample preparation: the jujube was first taken out of the refrigerator and cut into slices with a thickness of 0.5 ± 0.1 mm after the jujube temperature returned to room temperature. The slices were evenly laid on the acrylic plate in a single layer. The initial mass was recorded and then put into the oven, preheated for 30 min with a set air temperature of 60 °C and air speed of 2.5 m/s for drying. The drying is stopped when the sample is dried to the target moisture content. Then the sample was cut into circular slices with a diameter of 5.5 mm by a circular perforator after the sample was cooled to room temperature. Then the samples were put into the crucible and weighed with an analysis balance. The samples with a mass between 5 mg and 10 mg were selected and put in a crucible. A tablet press compacted the crucible with a cover to make the sample. Meanwhile, the empty crucible and the cover were compacted and made into standard samples for subsequent experiments.

After the main furnace was turned on and preheated for 30 min, the measuring program was set with the software, and the sample quality, crucible material, and other related information were filled in. The cover plate was opened, the standard and the sample were placed in the furnace, respectively, the sample was set to rise from 15 °C to 100 °C at a heating rate of 15 °C/min, and the relevant data was recorded. The test was repeated three times, and the average value was taken as the result.

#### 2.4.2. Thermal Conductivity

The thermal conductivity of jujube slices is an essential thermal characteristic parameter. In this study, the double-needle detection method was used to detect the thermal conductivity of jujube slices, which has the advantages of fast measurement speed, high accuracy, simple operation, and a wide application range. The thermal conductivity of the jujube slice under different temperatures, and moisture content was measured by the KD2Pro thermal characteristic analyzer.

Sample preparation: the jujube was first taken out of the refrigerator; when the temperature returned to room temperature, the jujube was cut into slices with a thickness of 8 ± 1 mm and uniformly laid on the material plate in a single layer. The dates were put into the oven with a set air temperature of 60 °C and an airspeed of 2.5 m/s and preheated for 30 min for drying, and the drying was stopped when the sample was dried to the target moisture content. 

The dried jujube slices were sealed and stored, and then heated to the target temperature in a moisture bath, sh3-type probe was inserted into the multi-layer jujube slices to ensure that there was no gap between the jujube slices, and the probe was completely inserted into the jujube slices, with the tail end not exposed to air. After that, the equipment started recording the thermal conductivity, temperature, and other information. The experiment was repeated three times, and the average value was taken as the result.

#### 2.4.3. Dielectric Properties

In this study, the dielectric properties of a jujube slice were measured by a terminal open-circuit coaxial probe dielectric characteristics measuring system. 

Furthermore, the jujube slices were punched into a cylinder with a diameter of 22 mm by using a punching tool, and the sample should be fully in contact with the fixture during the experiment process.

Before the measurement, the measurement system was calibrated according to the standard to ensure the accuracy of the test. Firstly, the impedance analyzer and the computer were turned on and preheated for 30 min. Furthermore, the calibration kit (Agilent E4991B-010, Santa Clara, CA, USA) was used to connect and calibrate the impedance analyzers, respectively, for open circuit, short circuit, low loss capacitance, and 50 Ω resistance. Air and 25 °C deionized moisture were used to calibrate the open-end coaxial probe of the equipment. To reduce mechanical errors, ensure that the impedance analyzer, coaxial cable, and probe are fixed during calibration and measurement.

The processed round jujube slices with a flat surface are placed in the sleeve when measuring. The lower part of the sleeve was sealed with a rubber plug fixed to the temperature sensor and placed on the hydraulic lifting platform to adjust the height of the hydraulic lifting platform to make sure that the surface of the jujube slices in the sleeve could fully contact the probe. Starting from room temperature, set the measuring temperature value, adjusted the circulating temperature control system, and gradually warmed up. When the temperature required by the required test is reached, the data of 101 frequency points within the logarithmic coordinate can be collected using the instrument’s measurement software (Keysight Materials Measurement Suite 2020) in the frequency range of 1~300 MH. Considering the processing temperature range of the jujube slice, five temperature points of 20 °C, 35 °C, 45 °C, 60 °C, and 80 °C were selected to measure the dielectric parameters of the jujube slices. Before measuring a new sample, the probe and fixture were cleaned with deionized moisture, dried with absorbent paper, and recalibrated. The experiment was repeated three times, and the average value was taken as the result.

#### 2.4.4. Effective Moisture Diffusion Coefficient

Diffusion is the main moisture movement mechanism in the drying process of agricultural products. In this study, assuming that moisture evaporation only occurred on the surface of the material and internal moisture was transferred to the surface of the material through diffusion, the Fick diffusion equation was used to describe the effective moisture diffusion coefficient of jujube slices under hot air and RF hot air combined drying conditions. This is consistent with Song Ruikai’s simulation hypothesis of microwave drying of potatoes [30]. The calculation formula is as follows (7):(7)$$\frac{\partial M_{t}^{db}}{\partial t}=\nabla \cdot (D_{eff}\nabla M_{t}^{db})$$
where *D_eff_* is the effective moisture diffusion coefficient.

The above equation can be solved by considering the sample’s geometry, initial drying conditions, and boundary conditions. In this study, jujube slices’ effective moisture diffusion coefficient was calculated under HA and HA + RF drying conditions. Assuming that jujube slices had uniform initial moisture distribution and a constant diffusion coefficient in the definite drying stage, ignoring sample shrinkage and external mass transfer resistance, the effective moisture diffusion coefficient of samples with different thicknesses can be expressed as follows (8) [31,32]:(8)$$MR=\frac{8}{\pi^{2}}-\exp(\frac{\pi^{2}D_{eff}}{4L^{2}}t)$$
where *L* is the thickness of the jujube slice and *MR* is the moisture ratio. The thickness of the jujube slice used in this experiment is 8 ± 0.5 mm. The calculation formula is as follows (9) [33]:(9)$$MR=\frac{M_{t}^{db}-M_{e}^{db}}{M_{0}^{db}-M_{e}^{db}}$$
where $M_{0}^{db}$ is the initial dry base moisture content of the sample (%), and $M_{e}^{db}$ is the dry base moisture content (%) when the sample is dried to equilibrium.

#### 2.4.5. Mass Transfer Coefficient

In order to calculate the mass transfer coefficient, lvan Shorstkii et al. obtained the mass transfer coefficient using the Luikov methodology based on the thermodynamic method [34]. The convective heat transfer coefficient and mass transfer coefficient calculated in this paper were based on the Dincer model, which has been used in many studies to estimate the mass transfer coefficient of agricultural and sideline products during volumetric heating [26,35,36]. This model was used to calculate the mass transfer coefficient, convective heat transfer coefficient, Biot number, and drying constant of jujube slices dried by single hot air and RF hot air combined drying.

Define the moisture ratio (*MR*) according to the dimensionless lag factor and drying coefficient, as shown in Equation (10) [20]:(10)$$MR=L_{f}\exp(-kt)$$

*L_f_* is a hysteresis factor (an infinite programmatic number); *k* is the drying coefficient. The experimental drying data *MR* Was fitted into Formula (10) to determine the values of *L_f_* and *k*. The dimensionless bidirectional number (*B_i_*) is calculated according to Formula (11):(11)$$L_{f}=\exp\left(\frac{0.2533\times B_{i}}{1.3+B_{i}}\right)$$

The estimated effective moisture diffusion coefficient is calculated by Formula (12):(12)$$D=\frac{k\times L^{2}}{\mu_{1}{}^{2}}$$
where *D* is effective moisture diffusion coefficient m^2^/s; *L* is half the thickness of the material (m), and *μ*_1_ is the dimensional root of the characteristic equation.

According to *B_i_*, calculate *μ*_1_ by Formula (13):(13)$$\left\{\begin{array}{ll} \mu_{1}=\frac{\pi }{2} & \left(B_{i}\geq 100\right)\\ \mu_{1}=\tan^{-1}(0.640443B_{i}+0.380397) & \left(B_{i}<100\right) \end{array}\right.\;$$

Calculate mass transfer coefficient hm (m/s) by combining Formulas (13) and (14) [18,37,38,39]: (14)$$h_{m}=\frac{B_{i}\times D}{L}$$

#### 2.4.6. Convective Heat Transfer Coefficient

The convective heat transfer coefficient *h_T_* (W/m^2^K) can be calculated from Formulas (15) to (18):(15)$$h_{T}=\frac{Nuk_{a}}{L}$$

Nusselt number (*Nu*) can be calculated by Formula (16):(16)$$Nu=0.664Re^{0.5}\;{\mathrm{Pr}}^{0.33}$$

Reynolds number (*Re*) and Prandtl number (Pr) can be calculated by Formulas (17) and (18) respectively:(17)$$Re=\frac{\rho_{a}vL}{\mu_{a}}$$
(18)$$\mathrm{Pr}=\frac{C_{pa}\mu_{a}}{k_{a}}$$
where *ρ_a_* is air density, (kg/m^3^); *μ_a_* is the dynamic viscosity of hot air, (Pa·s); *v* is the drying wind speed, (m/s); *L* is the feature length, (m); *C_pa_* is the specific heat of air, (J/kg·K); *k_a_* is the thermal conductivity of air, (W/m·K).

### 2.5. Development of Finite Element Model

#### 2.5.1. Development of Geometric Model

The geometric model of a single jujube slice is shown in Figure 4. A lower plate is provided at 20 mm on the lower surface of the jujube slice, and an upper plate is set at 110 mm from the lower plate. The opposite area of the upper and lower plate sizes is (80 × 43) cm.

#### 2.5.2. Model Assumption

Basic assumptions: Within all the domains in the model:(1)All two phases (liquid, solid) in the food sample were continuous media;(2)The jujube slice was a cylinder, and no morphological changes occured during drying;(3)The initial moisture content and initial temperature of jujube slices were uniform, and evaporation occurred on the upper surface of the sample;(4)The moisture is present only in liquid and vapor form;(5)Diffusion is the main mechanism by which moisture migrates from inside the material to the surface;(6)The jujube slice samples were isotropic;(7)The characteristics of jujube slice thermal, physical characteristics, and dielectric properties.

#### 2.5.3. Resolution of Simulation Model

The finite element analysis software COMSOL 6.0 (Zhongfa Technology Co., LTD., Wuhan, China) was used to establish the numerical models. The hot air and RF hot air combined drying process was simulated, and the results were visualized. In this study, electromagnetic heating, heat transfer, and mass transfer were coupled, and the MUMPS solver was selected in COMSOL 6.0 to solve the iterative equations. The relative tolerance was set to 0.1, the absolute tolerance was set to 0.001, the time step was set to 1 s, and the grid was set to “more fine” (the jujube slices model contains 15,940 domain elements, 1884 boundary elements, 136 edge elements, the minimum mass was 0.2509, and the average mass was 0.6598). The initial temperatures (air, jujube slices) in all domains in the simulation model were set to room temperature (22 °C). The cavity wall of the RF heating cavity is a completely shielded shell, so the tangential component of the electric field E was zero, and the lower plate was grounded so that the *V* of the lower plate was zero. The simulation process was performed on an AMDR74800 h PC (Windows 10 64-bit operating system, 4.0 GHz processing speed, 8 GB random access reader). Figure 5. Schematic diagram of finite element modeling and basic steps of solution.

### 2.6. Model Verification

Lay the prepared winter jujube slices evenly on the polypropylene tray in a single layer. According to the pre-experiment, convection drying temperature and an air speed were 60 °C and 2.5 m/s, respectively. Before the drying experiment, the drying equipment should be preheated for 30 min to achieve the desired temperature. The hot air drying process (4.5 h) was divided into three isochronous stages, and RF was introduced into each stage separately. According to the preliminary experiment, the pad spacing was 110 mm, and the drying test was completed after 4.5 h.

#### 2.6.1. Moisture Content Detection

During the experiment, the initial mass of the jujube slice was recorded as *W*_0._ Sample weights were measured every 30 min and recorded as *W*_t_. The moisture content of the jujube slice was calculated as follows. The experiment was repeated three times, and the average value was taken as the result. The calculation formula is as follows: (19) and (20): (19)$$M_{t}^{db}=\frac{W_{t}-W_{d}}{W_{d}}\times 100\%$$
(20)$$M_{t}^{wb}=\frac{W_{t}-W_{d}}{M_{t}}\times 100\%$$
where *W_t_* is the total mass of the sample at time *t* (g); *W*_𝑑_ is the mass of the dry matter of samples (g); $M_{t}^{db}$ is the dry base moisture content of jujube slice at time *t* (%); $M_{t}^{wb}$ is the wet base moisture content of jujube slice at time *t*.

#### 2.6.2. Nuclear Magnetic Signal Acquisition of Moisture of Jujube Slice

Five jujube slices were taken out every 30 min, sealed, and stored to determine the low-field NMR relaxation spectrum during the drying process. LMR-30, the low-field NMR detector, was used to acquire the relaxation signals of jujube slices. The center frequency of the instrument was 7 MHz, and the CPMG pulse sequence was used for data acquisition. The pulse sequence parameters were: echo interval TE = 200 us, echo number 40,000, waiting time TW = 6 s, and 16 times of accumulation (40 times or 80 times when the moisture content of the jujube slice is lower than 40%). The T2 spectrum was calculated using the BRD fast inversion algorithm. The inversion calculation was set as follows: the number of T2 points is 128, the logarithmic point arrangement method is used, the minimum value of T2 is 0.01 ms, and the maximum value of T2 is 10 s. During the experiment, the date slices were wrapped in plastic wrap, put into the bottom of the test tube, and then kept in the thermostat for about half an hour. Subsequently, the test tube was placed into the sample chamber, the echo train signal was collected with the CPMG sequence, and the T2 spectrum was calculated after the scanning.

#### 2.6.3. Center Temperature Detection

Before the experiment, the probe of the optical fiber thermometer was inserted into the center of the jujube slices, and the temperature change of the jujube slices during the drying process was recorded in real-time by the optical fiber temperature measuring system.

#### 2.6.4. Surface Temperature Detection

During the drying process, the jujube slice’s surface temperature was obtained using an infrared thermal imager at intervals of 1.5 h. During the measurement, the tray was quickly removed from the heating chamber and placed in the field of view of the infrared camera to take pictures and obtain the surface temperature of the jujube slice. After shooting, quickly put the tray back into the heating chamber. In order to reduce the error, the whole process is controlled within 10 s.

## 3. Results and Discussion

### 3.1. Specific Heat Capacity Analysis

As shown in Figure 6, With the increase in temperature and moisture content, the specific heat capacity of jujube slices increases. Under the same moisture content (less than 60%), the specific heat capacity and its change rate of date slices increase and decrease with the increase in temperature, respectively. When the moisture content of jujube slices is greater than 60%, the specific heat capacity of jujube slices increases obviously with the increase in temperature, and the rate of change also shows a trend of increasing with the increase in temperature. Under the same temperature conditions, the higher the moisture content of jujube slices, the larger the specific heat capacity value. This is similar to the conclusions of Jiang’s research on the specific heat capacity of white radishes [29]. The regression equation representing the relationship between the specific heat capacity and temperature and moisture content of jujube slices was established as follows:(21)$$C_{p}=0.043T+1.047W-1.333$$

### 3.2. Thermal Conductivity Analysis

The change in thermal conductivity of jujube slices with temperature and moisture content is shown in Figure 7. It can be seen from the figure that the thermal conductivity of jujube slices also indicates a trend of increasing with an increase in temperature and moisture content. Under the same temperature condition, the jujube slice’s thermal conductivity is positively correlated with moisture content, while the change rate is approximately unchanged. The thermal conductivity of the jujube slice was also positively correlated with temperature. The regression equation representing the trend of the thermal conductivity and, temperature, and moisture content of the jujube slice was established:(22)$$k_{p}=\;0.008T+0.675W-0.261$$

### 3.3. Dielectric Properties Analysis

The dielectric constant of a jujube slice varies with temperature and moisture content at the RF frequency of 27.12 MHz, as shown in Figure 8. It can be seen from the figure that the permittivity of the jujube slice increases with the increase in temperature and moisture content. When the moisture content is constant and less than 40%, the permittivity of jujube slices increases significantly with the increase in temperature; however, when the moisture content of jujube slices is greater than 40%, the permittivity of jujube slices is above 70, close to that of pure moisture, and there is no noticeable change. At the same temperature, the permittivity of the jujube slice was positively correlated with the moisture content.

The regression equation representing the change trend of the dielectric constant and temperature and moisture content of the jujube slice was established:(23)ε′=34.519W+0.072T+46.022

The dielectric loss factor of the jujube slice at an RF frequency of 27.12 MHz of temperature and moisture content is shown in Figure 9 below. It can be seen from the figure that the dielectric loss factor of jujube slices increases with an increase in temperature and moisture content. Under the condition of the same moisture content, the dielectric loss factor of the jujube slice rises with the increase in temperature, and its change rate decreases with the increase in temperature. Under the same temperature condition, the dielectric loss factor of the jujube slice increases with the increase in moisture content, and its change rate also gradually decreases with the increase in temperature. The measured dielectric loss factor was used to do binary regression analysis on the temperature and moisture content of jujube slices, and the regression equation characterizing the changing trend of temperature and moisture content of jujube slices was established as follows:(24)$$\varepsilon^{\prime \prime }=\left(319.028\right)W+\left(1.150\right)T+\left(3.786\right)WT+46.952$$

### 3.4. Moisture Analysis

The following Figure 10 shows the comparison of experimental and simulated moisture content changes of jujube slices under hot air drying and RF-assisted hot air drying. It can be seen from the figure that the simulated results of moisture content (dry basis) are in good agreement with the test results. The determination coefficient (R^2^) between the simulated experimental values of HA, E-HA + RF, M-HA + RF, and L-HA + RF groups was 0.964, 0.987, 0.961, and 0.977, respectively. Both the simulation and experimental results show that the moisture content of the jujube slices decreases rapidly in the early stage of hot air drying, and the decreasing speed of the moisture content gradually decreases after 1.5 h. However, the decreasing speed of the moisture content of the jujube slices will be significantly improved after the addition of RF. Compared with the hot air-dried jujube slices at the same drying stage, the moisture content of the jujube slices dried by RF hot air combined drying showed a more obvious decreasing trend. The moisture content of the jujube slice was lower than that of the hot air drying at the same stage, thus shortening the drying time. Especially when RF is added in the middle and later stages of the hot air drying, the moisture content of the jujube slices will decrease more significantly, so that the moisture content of the jujube slices will quickly fall to the target moisture content. This may be because RF drying has the characteristics of volume heating, and the internal temperature of the experimental jujube slice increased rapidly, which greatly promoted the internal moisture migration to the outside, thus shortening the drying time. This result is also similar to the results of Zhou Xu’s RF hot air combined drying of kiwifruit slices [40] and Zhang et al.’s RF hot air combined drying of mango slices [12].

### 3.5. Nuclear Magnetic Signal Analysis

The moisture in jujube slices can be divided into free moisture, semi-combined moisture, and combined moisture, and their contents decrease in turn. Therefore, the low-field NMR T2 relaxation curve of moisture in jujube slices has three peaks, each representing three forms of moisture, and the peak area represents the moisture content in this form. According to Figure 11, it can be seen that with the drying process, the T2 relaxation peak of jujube slices moves to the lower left as a whole, the transverse relaxation time is shortened, the peak value becomes lower, and the peak area becomes smaller, which is caused by the evaporation and morphological changes of moisture in the drying process. In the hot air drying process, it is difficult to remove the moisture in the late drying period (3–4.5 h), so the relaxation peak of the corresponding time period is larger. After the addition of RF, the relaxation peak value and area were smaller, and the relaxation time was shorter than that of the hot air drying date slices at the same period, which further proves that the application of RF promotes the evaporation of moisture. The drying process was accelerated, and the drying time was shortened, which is in line with the experimental results shown in Figure 10.

### 3.6. Surface Temperature

The experimental results and simulation results of the surface temperature field of jujube slices in hot air drying and RF-assisted hot air drying are shown in Figure 12 below. It can be seen from the figure that the simulation results are in good agreement with the experimental results, with a maximum difference of 1.77 °C and a minimum difference of 0.3 °C. Both the simulation results and the experimental results showed that the surface temperature of the jujube slices increased gradually and gradually approached the temperature of the hot air in the process of hot air drying. In contrast, the surface temperature of the jujube slices was higher than that of the hot air drying in the same stage, which may be related to the characteristics of volume heating of the radio frequency. The whole temperature of jujube slices increased due to the vibration friction of moisture molecules when RF was added, which was reflected in the simulation and experimental results. According to the experimental values, compared with the hot air drying jujube slices in the same period, the surface temperature of the E-HA + RF group increased from 47.1 °C to 57.5 °C, the average surface temperature of the M-HA + RF group increased from 52.1 °C to 71.1 °C, and the average surface temperature of the L-HA + RF group increased from 54.2 °C to 74.9 °C.

### 3.7. Center Temperature

The experimental and simulated values of the central temperature of jujube slices in hot air drying and RF-assisted hot air drying are shown in Figure 13 below. The maximum temperature difference between the two is 6.9 °C, and the minimum is 0.01 °C. As can be seen from the figure, in the process of hot air drying, the central temperature of the jujube slice increased gradually, and the heating rate decreased gradually with time. Even at the end of drying, the central temperature of the jujube slice is still lower than that of the hot air. However, when the RF was applied, the central temperature of the jujube slice increased significantly and rapidly. This is because hot air drying transfers heat to the surface of the jujube slices through convection heat transfer and then to the inside of the jujube slices through heat conduction, with low heat transfer efficiency and a slow temperature rise. At the same time, radio frequency directly acts on the inside of the jujube slices and has the characteristic of “selective heating.” That is, the friction of moisture molecules in the central part of the jujube slices with high moisture content is more intense. As time goes by, the central temperature of jujube slices will gradually rise. The simulation result is similar to the research on radiofrequency heating of potato cubes [41]. After the RF application, the central temperature of jujube slices will drop rapidly, and the later the application stage, the higher the central temperature of jujube slices will rise.

## 4. Conclusions

In this study, we used COMSOL 6.0 Multiphysics software to build a simulation model of mass heat transfer coupling of jujube slices under HA and HA + RF drying conditions. After testing and verification, the characteristics of the temperature field and moisture field distribution of hot air drying and RF-assisted hot air drying were analyzed. The specific conclusions are as follows: (1) The theoretical-empirical mathematical model can characterize the mass heat transfer in the drying process of date slices. (2) The temperature-moisture physical field coupling model based on COMSOL 6.0 Multiphysics software can realize the simulation of jujube drying under the conditions of hot air drying and radio frequency-assisted hot air combined drying. There is good consistency between the simulation results and the test results. (3) Through simulation and test results, it can be seen that radio frequency impacts the temperature distribution and moisture migration of hot air-dried jujube slices. After adding radio frequency, the surface temperature and center temperature of jujube slices have increased significantly and rapidly, with the center temperature being higher than the surface temperature. The moisture content of jujube slices after adding radio frequency is faster than that of hot air-dried jujube slices in the same period. The later the stage of adding radio frequency, the more pronounced the temperature rise of jujube slices, and the faster the moisture content decreases.

## Figures and Tables

**Figure 1 foods-12-03025-f001:**
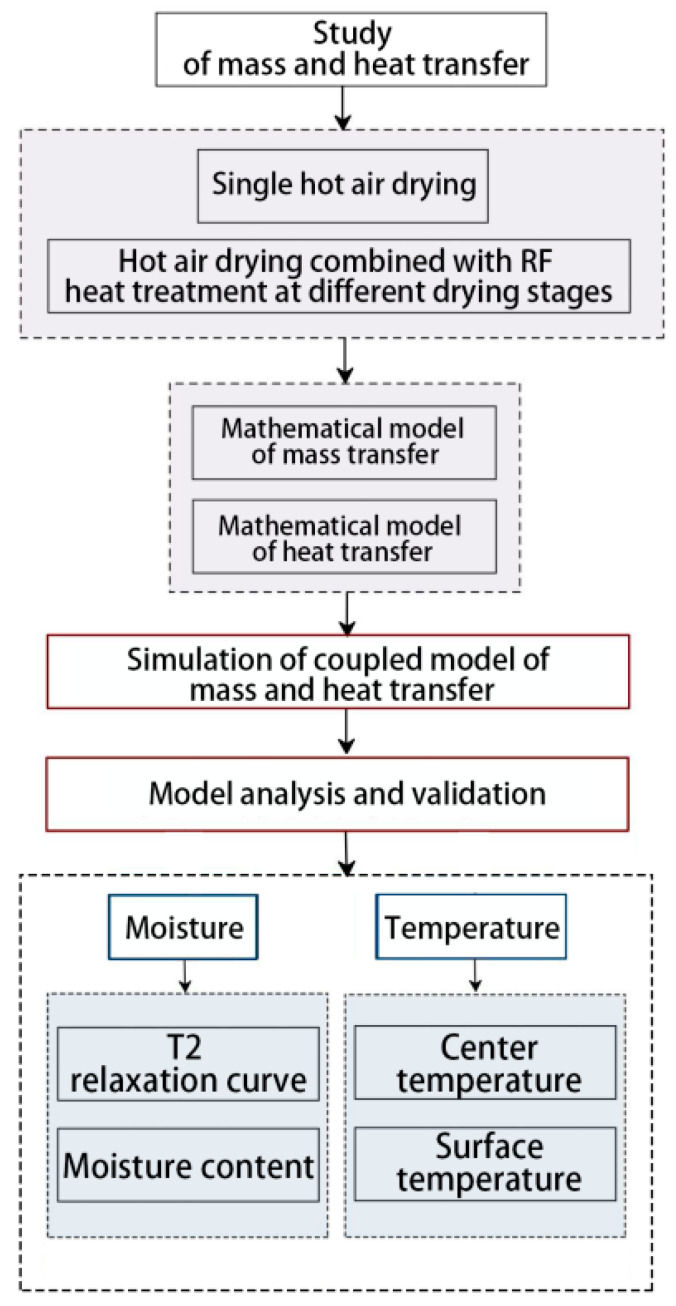
Flow chart of mass heat transfer research.

**Figure 2 foods-12-03025-f002:**
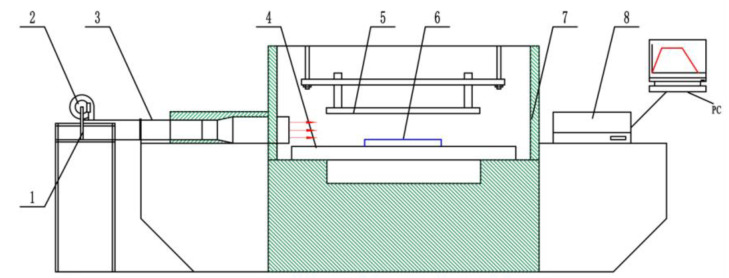
RF-hot air combined drying system **1**. Heating pipe **2**. Centrifugal fan **3**. Metal ventilation ducts **4**. Lower plate **5**. Upper plate **6**. Materials tray **7**. Rf equipment rack **8**. Optical fiber temperature measurement system.

**Figure 3 foods-12-03025-f003:**
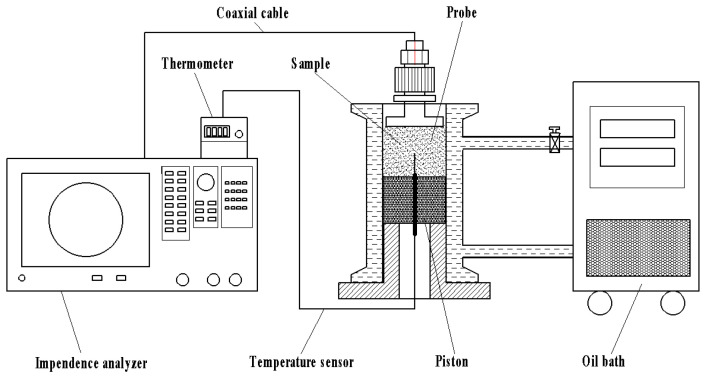
Terminal open-circuit coaxial probe dielectric characteristics measurement system.

**Figure 4 foods-12-03025-f004:**
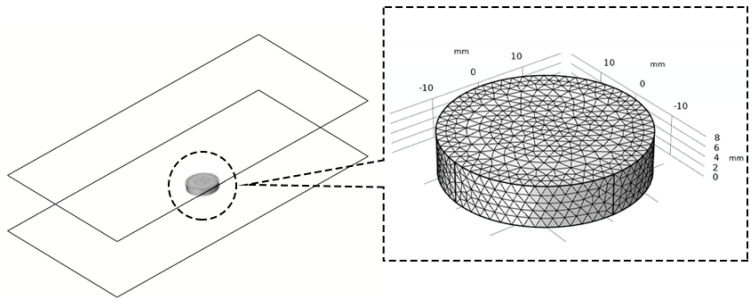
Geometric model of jujube slice.

**Figure 5 foods-12-03025-f005:**
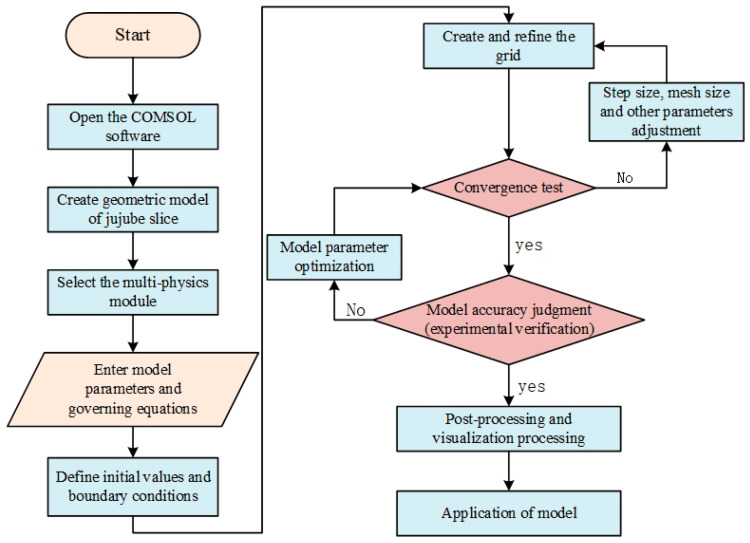
Schematic diagram of finite element modeling and basic steps of solution.

**Figure 6 foods-12-03025-f006:**
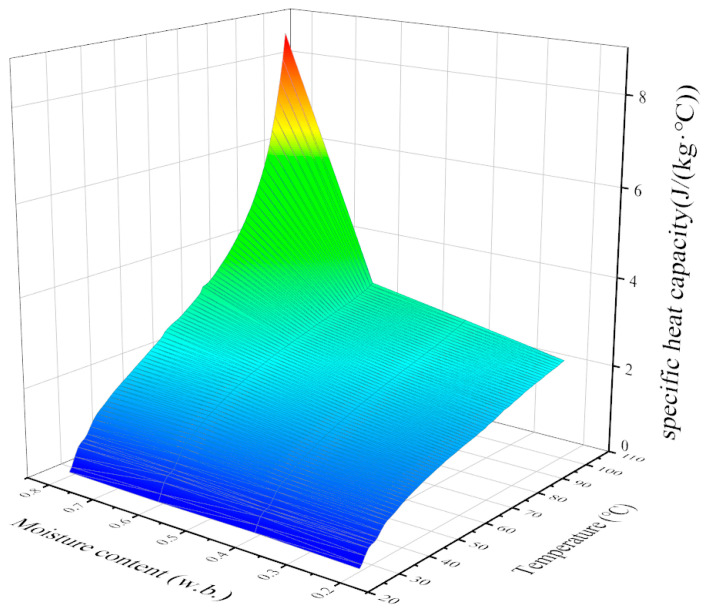
Relationship between specific heat capacity and temperature and moisture content of jujube slice.

**Figure 7 foods-12-03025-f007:**
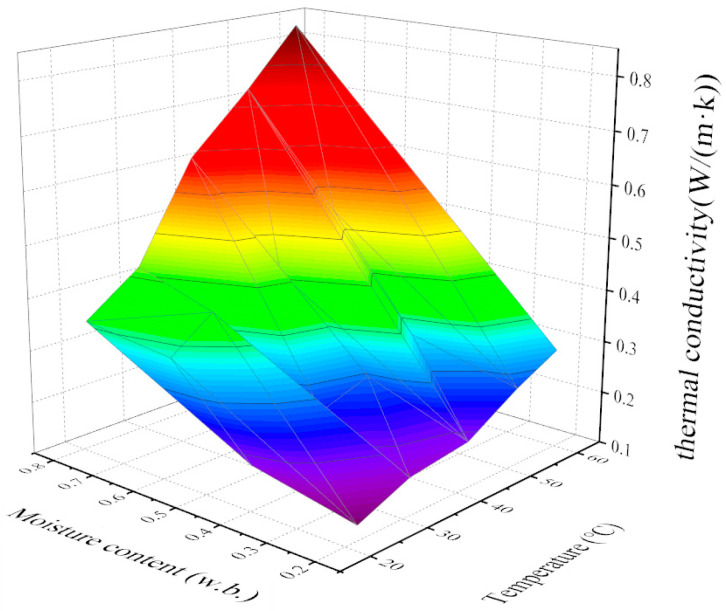
Relationship between thermal conductivity and temperature and moisture content of jujube slice.

**Figure 8 foods-12-03025-f008:**
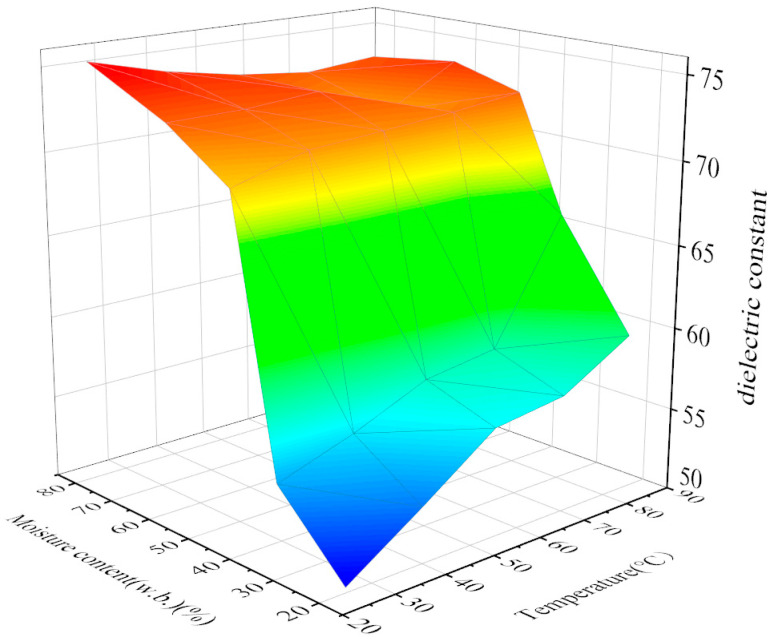
Relationship between dielectric constant and temperature and moisture content of jujube slice.

**Figure 9 foods-12-03025-f009:**
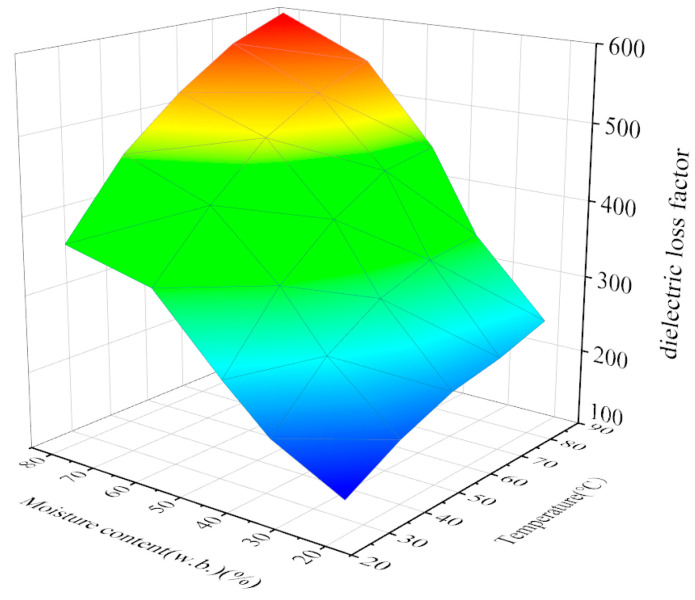
Relationship between dielectric loss factor and temperature and moisture content of jujube slice.

**Figure 10 foods-12-03025-f010:**
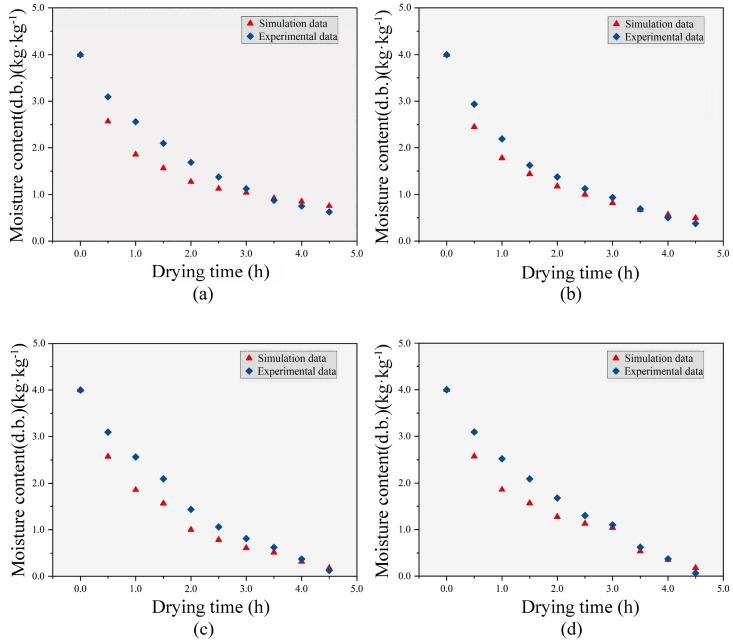
Comparison of dry base moisture content between experimental value and simulated value in the drying process of jujube slices (**a**) HA; (**b**) E-HA + RF; (**c**) M-HA + RF; (**d**) L-HA + RF.

**Figure 11 foods-12-03025-f011:**
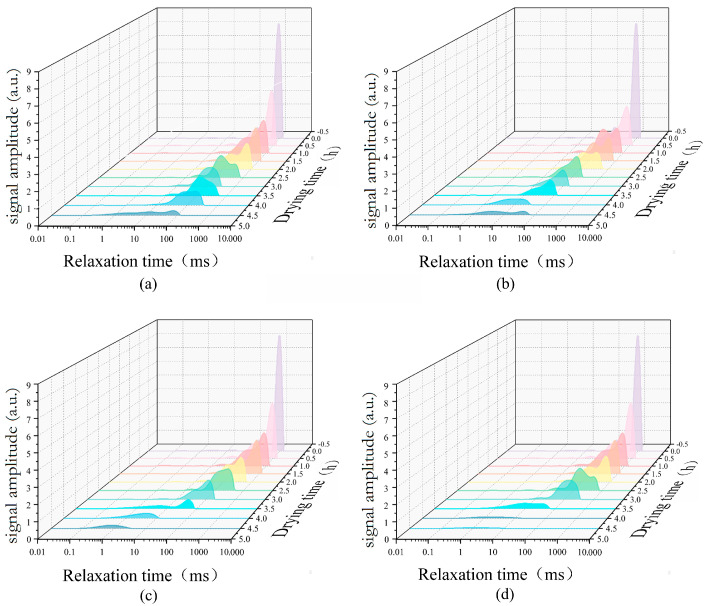
T2 relaxation curve of low field NMR during drying of jujube slices (**a**) HA; (**b**) E-HA + RF; (**c**) M-HA + RF; (**d**) L-HA + RF.

**Figure 12 foods-12-03025-f012:**
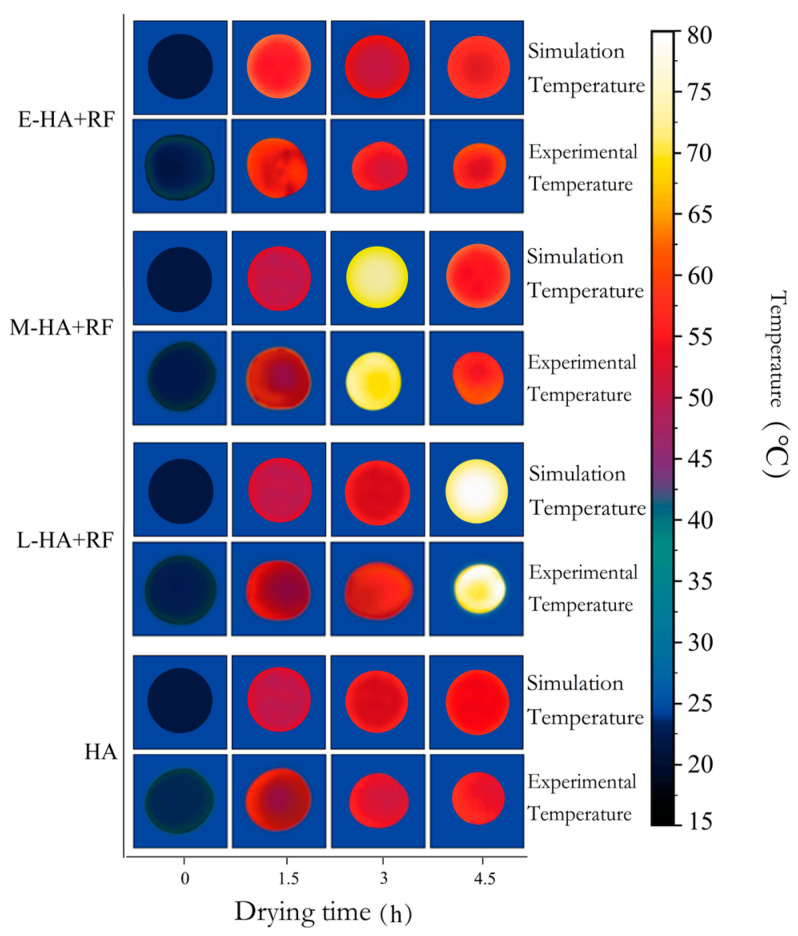
Experiment and simulated surface temperature profiles of jujube slice during drying process.

**Figure 13 foods-12-03025-f013:**
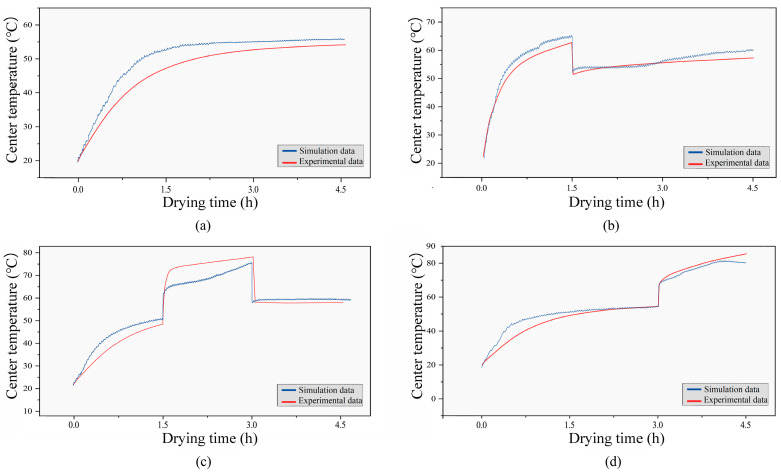
Experiment and simulated center temperature profiles of jujube slice during drying process (**a**) HA; (**b**) E-HA + RF; (**c**) M-HA + RF; (**d**) L-HA + RF.

## Data Availability

The data presented in this study are available on request from the corresponding author.

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
