# Peer review of "Simulated and Verification of Mass and Heat Transfer Coupled Model of Jujube Slices Dried by Hot Air Combined with Radio Frequency Heat Treatment at Different Drying Stages"

_foods, 2023, doi:10.3390/foods12163025_

Round 1
Reviewer 1 Report
This is an interesting paper that assesses the impact of Hot Air Combined with Radio Frequency Heat Treatment on Mass and Heat Transfer Coupled Model of Jujube Slices. It is well organized and the results are interesting. While I have many comments for improving readability and emphasizing key results, the manuscript is worth consideration after addressing these comments and revising the manuscript.
1. Abstract: The authors stated: The determination coefficient (R2) between the simulated experimental value of HA and E-HA+RF, MHA+RF, L-HA+RF group was 0.964, 0.987, 0.961, and 0.977 respectively. Please clarify what E, M and L means in this groups.
2. Introduction. RF is a highly intensive process. Why do authors did not states any influence of a pressure potential. The temperature of the surface and in the center of the drying product are extremely different.
3. M&M. Section 2.2. The major parameter for RF is frequency. I did not find any information regarding frequency value.
4. M&M section 2.3.1. The pressure gradient is a major force of such intensive mass transfer. Thus pressure can not be neglected. Please add an explanation of used model. Additionally, please add comments on thermodiffusion, convective diffusion etc.
5. M&M section. Temperature measurement. What temperature was obtained from fiber optic thermometer? Surface temperature? Any methods to measure an internal temperatre?
6. M&M. Section 2.4.3. Which part of jujube slice was used to obtain the dielectric properties?
7. M&M. Section 2.4.5. The authors stated: The Convective heat transfer coefficient and mass transfer coefficient calculated in this paper were based on Dincer model. Please add citation. Based on thermodynamical approaches the mass transfer coefficient can be find using Luikov methodology (https://doi.org/10.1016/j.icheatmasstransfer.2020.105003 or https://doi.org/10.1016/j.jfoodeng.2021.110822) Please add comments of the necessity of using the Dincer model.
8. M&M. Section 2.4.4. In which conditions the Effective water diffusion coefficient was obtained? HA? Or during combination of suggested methods? What was the jujube thickness? Normally. The equation (8) should be used if (diameter/thickness)>10.
9. M&M section 2.5.2. The model assumptions for suggested model are poor: the authors should add in which form moisture presents in the drying material, chemical reactions? Molar capillary transfer?
Based on the above points, I would propose a major revision of the manuscript.
Reviewer 2 Report
There are some serious issues with Equations used in this paper.
They have mentioned developed a new model.
But it has not been discussed anywhere in the article.
The article was based on drying the material using RF. The authors discussed the technical issues why the heat transfer through RF is better than the conventional techniques.
My main concern was the language of this paper and error in the computational work.
I think it is concise and short feedback of the submitted article.
Some sentences do not make any sense.
Reviewer 3 Report
The topic of the manuscript is interesting.
The Authors explain the challenge of non-uniform heating but another challenge which is relevant is non-uniform water content within the product. Indeed, the moisture content of the product affects its water activity, which in turn is related to the rate of oxidative reaction and non-enzymatic browning. Indeed, studies on intermediate moisture and dried products as a function of moisture content or water activity have allowed the development of mathematical models that predict changes in product chemical or physical properties over time as a function of water activity (see: Corey, M. E., Kerr, W. L., Mulligan, J. H., Lavelli, V. Phytochemical stability in dried apple and green tea functional products as related to moisture properties. LWT - Food Sci. Technol., 2011, 44, 67-74). There are significant differences in the rate of chemical reactions as a function of water activity and hence it is important that during drying the products or parts of the products if moisture is not distributed uniformly, do not remain at the most critical water content values for long times. Hence a tool that predicts the moisture and temperature level in a food product during drying is advisable.
In general, I suggest discussing the impact of non-uniform temperature a non-uniform water distribution on product quality in order to justify the need of the predictive tool.
Round 2
Reviewer 1 Report
The authors answered my comments in current manner. The manuscript now can be accepted.
Reviewer 2 Report
It can be considered.
NA